# FEATURE SELECTION IN THE PRESENCE OF MONOTONE BATCH EFFECTS

## ABSTRACT

We study the feature selection problem in the presence of monotone batch effects when merging datasets from disparate technologies and different environments affects the underlying causal dependence of data features. We propose two novel algorithms for this task: 1) joint feature selection and batch effect correction through non-linear transformations matching the distribution of data batches; 2) transforming data using a batch-invariant characteristic (i.e., feature rank) to append datasets. To match the distribution of data batches during the feature selection procedure, we use the maximum mean discrepancy (MMD) distance. We assess the performance of the feature selection methods used in conjunction with our batch effect removal methods.

Our experiments on synthetic data show that the former method combined with Lasso improves the $F_1$ score significantly, even with few samples per dataset. This method outperforms popular batch effect removal algorithms, including Combat-Seq, Limma, and PCA. Comparatively, while the ranking method is computationally more efficient, its performance is worse due to the information loss resulting from ignoring the magnitude of data.

## 1 INTRODUCTION

Public-use datasets are becoming increasingly popular as funding requirements, data transparency, and open-source culture push researchers to share gathered data. Aggregation of related datasets allows for greater statistical power during analysis, particularly in the biological and genomics data contexts, where each experiment may only contain a few samples due to high experimental costs.

Merging datasets from disparate environments comes with challenges, as datasets from the same environment may be subject to similar biases. For example, differences in genome sequencing machines Pareek et al. (2011), hybridization protocols Young et al. (2020), and transformation methods Robinson & Oshlack (2010); Risso et al. (2014) may lead to batch effects (i.e. systematic non-biological differences between batches of samples) in gene expression data. Batch effects can harm the performance of statistical inference algorithms (in particular, feature selection for detecting useful biomarkers) by imposing bias on the predictions, increasing the false discovery rate, and reducing prediction accuracy Sims et al. (2008). Thus, detection and removal of batch effects are crucial pre-processing stages of the statistical analysis of various bioinformatics tasks such as (single-cell) RNA sequencing Chen et al. (2011), metabolomics analysis Liu et al. (2020), and cancer classification Almugren & Alshamlan (2019); Leek et al. (2010).

Prior literature has studied the problem of mitigating batch effects. These methods can be categorized into several general categories. Clustering and KNN-based methods remove batch effects by finding the common sources of variations among datasets based on the proximity of data points (Butler et al., 2018; Zhang et al., 2019; Li et al., 2020; Fang et al., 2021; Lakkis et al., 2021). More specifically, each data batch is considered a cluster, and the batch effect is viewed as the between-cluster variances Fang et al. (2021). These methods are computationally expensive due to computing pairwise distances of data points. Further, they might not perform well if the batch effect changes the original distances of data points drastically and differently among different batches.

Another collection of methods is based on reducing the dimension of the original data by removing the batch effects as spurious dimensions in data (Haghverdi et al., 2018; Alter et al., 2000; Butler et al., 2018). More specifically, they apply a pre-processing algorithm such as t-SNE Zhang et al. (2019), Principle Component Analysis Haghverdi et al. (2018), Singular Value Decomposition Alter et al.

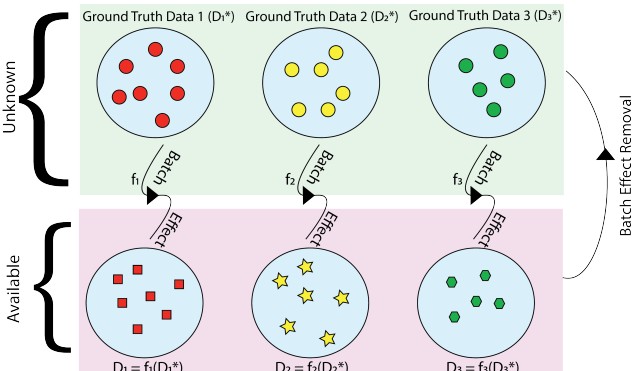

Figure 1: The batch effect can be conceptualized as transformations on each of the ground-truth datasets, which changes the distribution of data in each in a potentially different way.

(2000), UMAP Tran et al. (2020), or Canonical Correlation Analysis Butler et al. (2018) to project the data into a lower dimensional sub-space. Then, the noise components (results of the batch effect) are removed from the signal data (ground truth data which is free of the batch effect). A common disadvantage of such methods is that by projecting different batches onto the same low-dimensional space, valuable information contributing to the inference phase may be lost. An alternative approach is to apply a clustering algorithm on the data and remove the batch effect iteratively through the clustering procedure Li et al. (2020); Fang et al. (2021); Lakkis et al. (2021). More specifically, each data batch is considered a cluster, and the batch effect is viewed as the between-cluster variances Fang et al. (2021). A common disadvantage of such methods is that by projecting different batches onto the same low-dimensional space, valuable information contributing to the inference phase may be lost.

Yet another class of approaches formulate the batch effect problem as a parametric model and estimate the unknown parameters by classification or regression techniques Johnson et al. (2007); Lazar et al. (2013); Leek (2014); Risso et al. (2014); Robinson et al. (2010); Love et al. (2014); Zhang et al. (2020). The most popular method in this category is arguably ComBat-seq Zhang et al. (2020), which considers a joint negative binomial regression model for gene counts and the batch effects. Their approach adjusts the model's parameters by matching the cumulative density functions of the data generated by the model and the original data. The common problem of these approaches is the strict parametric assumptions on the data distribution (such as negative binomial) and the model (linearity) that limit their applicability. Finally, the aforementioned methods apply batch effect removal and inference tasks on data in separate phases. Thus, the error in the batch effect removal phase can be propagated to the inference phase.

A more recent class of methods removes the batch effect by matching the empirical distribution of different batches by minimizing a distribution distance measure such as Kullback-Leibler (KL) Divergence Lakkis et al. (2021) or maximum mean discrepancy (MMD) Shaham et al. (2017); Niu et al. (2022). Shaham et al. (2017) mitigates batch effect by minimizing the Maximum Mean Discrepancy (MMD) between the two distributions of samples. They consider one dataset as the reference, and the other dataset is transformed to have a similar distribution as the reference dataset. While their algorithm works for two input datasets, our methodology utilizes all input datasets to simultaneously reduce the batch effect on all of them. Another significant difference between our framework and theirs is that we perform joint batch removal and feature selection together instead of doing these steps independently. As discussed, removing batch effects jointly with feature selection can significantly enhance the feature selection accuracy. The reason is many feature selection strategies (such as the popular LASSO algorithm) make certain assumptions on the dataset (e.g., generalized linear models). Such assumptions may be invalidated when the batch effect removal procedure does not account for the downstream feature selection procedure.

**Feature Selection Methods:** Feature selection when no batch effect is involved is extensively studied in the literature (Saeys et al., 2007; Jović et al., 2015). A popular approach selects features with the highest correlation (e.g., Pearson or Spearman (He & Yu, 2010)) with the target variable. To choose the most relevant features, one can then compute the p-value for each computed correlation and choose the ones with the highest correlation after adjusting with false discovery rate control mechanisms such as the Benjamini-Hochberg procedure (Benjamini & Hochberg, 1995). Notice

that one can rely on exact statistical approaches such as permutation tests to avoid distributional assumptions. A more popular approach for finding the most relevant features with the highest prediction power of the target variable is to formulate the problem as a Lasso regression task. The method's output is the features corresponding to the non-zero elements of the regression parameter vector Vinga (2021).

In this work, we jointly handle the feature selection task and batch effect removal through data transformation. In particular, we remove the batch effect by finding the optimal transformations that minimize the maximum mean discrepancy (MMD) of different data batches.

## 2  PROBLEM FORMULATION AND METHODOLOGY

Let us first rigorously define the problem of distribution-free feature selection in the presence of batch effects: let $\mathcal{D}_1, \ldots, \mathcal{D}_m$ be a collection of datasets from $m$ unique laboratories studying the interactions of $d$ input features (e.g., gene expressions, proteomics data, etc.) and a target variable $y$. Ideally, all datasets in the collection follow the identical distribution $P^*$ describing the joint distribution of input features and the target variable. However, due to the aforementioned factors in the previous section, known as the batch effect, the datasets in the collection do not follow the ground-truth distribution $P^*$. Formally, the batch effect can be described as $m$ different transformations $f_1, \ldots, f_m$ applied to the ground-truth datasets $D_1^*, \ldots, D_m^* \sim P^*$ (see Figure 1). **We assume that all functions $f_i, i = 1, \ldots, m$, are monotonically increasing.** Such an assumption on the monotonicity of batch effects holds for the widely used transformations on the raw microarray and sequencing datasets, such as the log transformations and dividing by the maximum values. Thus each observed dataset can be viewed as a monotone transformation of the ground-truth data:

$$\mathcal{D}_i = f_i(\mathcal{D}_i^*) \quad i = 1, \ldots, m. \tag{1}$$

The goal of batch effect removal is to learn the underlying transformations $f_1, \ldots, f_m$ on the data batches such that the ground-truth datasets $\mathcal{D}_1, \ldots, \mathcal{D}_m$ are recovered up to bijective mappings (see Figure 1). In other words, the optimal transformations on the datasets make the distributions of them as *close* as possible to each other. Thus, one can quantify the quality of batch effect removal based on how close the resulting distributions of datasets are after the transformations. Note that it is crucial for transformations to be bijective. Otherwise, one can transform all datasets to zero, and the difference between the transformed datasets will be zero. However, such transformations are not desirable. Unfortunately, finding the optimal transformations over the set of bijective functions is challenging from the optimization point of view. We propose a novel formulation in the next section to avoid such spurious solutions.

### 2.1  MMD-BASED APPROACH

We utilize the maximum mean discrepancy Gretton et al. (2012) to measure the proximity of dataset distributions Arjovsky et al. (2017). Maximum Mean Discrepancy (MMD) is a statistical measure that quantifies the dissimilarity between probability distributions. Given two datasets or distributions, MMD aims to determine the dissimilarity by calculating the discrepancy between their respective means in the Reproducing Kernel Hilbert Space (RKHS). Mathematically, let $z$ and $z'$ be independent random variables following distribution $p$, and $w$ as well as $w'$ independent random variables following distribution $q$. Then, the MMD between distributions $p$ and $q$ is

$$\text{MMD}^2[p, q] = \mathbf{E}_{z,z'}\left[k\left(z, z'\right)\right] - 2\mathbf{E}_{z,w}[k(z, w)] + \mathbf{E}_{w,w'}\left[k\left(w, w'\right)\right] \tag{2}$$

An empirical estimate of MMD is given by (Gretton et al., 2006)

$$\text{MMD}_u^2[x, y] = \frac{1}{m(m-1)} \sum_{i=1}^{m} \sum_{j \neq i}^{m} k\left(x_i, x_j\right) + \frac{1}{n(n-1)} \sum_{i=1}^{n} \sum_{j \neq i}^{n} k\left(y_i, y_j\right) - \frac{2}{mn} \sum_{i=1}^{m} \sum_{j=1}^{n} k\left(x_i, y_j\right),$$

where $k(\cdot, \cdot)$ is a non linear kernel function. In our work, we apply the Gaussian kernel, defined as

$$k(z, w) = \exp\left(\frac{-|z - w|^2}{2\sigma^2}\right).$$

Here, $\sigma$ is a hyper-parameter to control the bandwidth of the kernel. To accurately estimate MMD and avoid the vanishing gradient phenomenon, we use multiple kernels with different bandwidths taking values in $\sigma = [10^{-2}, 10^{-1}, \ldots, 10^{11}]$, which is proper for even large distributional differences.

We assume the input features $\mathbf{x}^*$ without batch effect follow a multi-variate Gaussian distribution $\mathcal{N}(\mu, \Sigma)$ with the zero mean and an unknown covariance matrix. Therefore, we can formulate the batch effect removal problem as finding the optimal transformation and covariance matrix $\Sigma$ such that the total difference between transformed distributions and the underlying normal distribution is minimized:

$$\min_{\boldsymbol{\Phi}, \Sigma} \sum_{i \in E} \text{MMD}\Big(\boldsymbol{\Phi}_i(\mathcal{D}_i), \mathcal{N}(0, \Sigma)\Big), \tag{3}$$

where $\text{MMD}(\cdot, \cdot)$ measures the maximum mean discrepancy between the two input distributions/datasets. A two-layer neural network models each $\boldsymbol{\Phi}_i$ transformation. We add non-negativity constraints on the weights of the neural networks to ensure the corresponding transformations are monotone:

$$\Phi(\mathcal{D}_i) = \mathbf{U}^T \text{ReLU}\Big(\mathbf{W}_2 \text{ReLU}(\mathbf{W}_1 \mathbf{X}_i + \mathbf{b}_1) + \mathbf{b}_2\Big) + \mathbf{c} \tag{4}$$

Where $\mathbf{W}_1, \mathbf{W}_2, \mathbf{U}, \mathbf{b}_1, \mathbf{b}_2,$ and $\mathbf{c}$ are the neural network trainable weights with the non-negativity constraints. Further, $\mathbf{X}_i$ represents the data matrix associated with the dataset $\mathcal{D}_i$. One can argue that instead of transforming the data to a common normal distribution, we can minimize the pairwise MMD distances of the distributions. However, in this case, transforming all distributions to a constant value such as $0$ is the optimal solution with $0$ loss, which is clearly undesirable. An alternative approach used by Shaham et al. (2017) takes one of the dataset distributions as the "reference" and then it brings the distribution of other datasets to the chosen reference dataset. This approach can lead to models with poor performances if the batch effect highly affects the reference data. Our experiments show that even when we randomly generate the covariance matrix $\Sigma$ instead of considering it as an optimization parameter, still it achieves a better performance compared to the case when one of the data distributions considered as the reference.

One can optimize Problem (3) to obtain the transformations removing batch effects in a pre-processing stage. Next, a feature selection method on the transformed data can be performed to find the set of input features that are most relevant for predicting the target variable. A more elegant approach is to remove batch effects and perform the feature selection in a joint optimization task. Note that performing these tasks separately and sequentially can lead to sub-optimal solutions compared to the joint formulation due to the data processing inequality (Beaudry & Renner, 2011). The joint batch effect removal via MMD and Lasso regression can be formulated as follows:

$$\min_{\boldsymbol{\Phi}, \boldsymbol{\theta}, \Sigma} \quad \frac{1}{n} \sum_{i=1}^{n} \sum_{(\mathbf{x}, y) \in \mathcal{D}_i} (\boldsymbol{\theta}^T \Phi_i(\mathbf{x}) - y)^2 + \lambda \|\boldsymbol{\theta}\|_1 + \mu \sum_{i=1}^{m} \text{MMD}\Big(\Phi_i(\mathcal{D}_i), \mathcal{N}(0, \Sigma)\Big). \tag{5}$$

The objective function consists of three main components: the first term represents the linear regression loss on the transformed dataset. The second part is the non-smooth $\ell_1$ regularizer controlling the sparsity of the model's parameter $\boldsymbol{\theta}$. Finally, the third component transforms the distribution of datasets to a common normal distribution. Problem (5) can be optimized using a first-order iterative algorithm where at each iteration, a step of projected gradient descent is applied on the parameters of the transformations. Then $\boldsymbol{\theta}$ is updated by the proximal gradient method. The procedure of optimizing (5) is presented in Algorithm 1.

---

**Algorithm 1** MMD-based feature selection and batch effect removal

---

1: Initialize $\boldsymbol{\theta}$ with normal distribution and $\boldsymbol{\Phi}_i$ randomly for all $1 \leq i \leq m$.
2: **for** $t = 1, \ldots, T$ **do**
3:     Update the parameters in $\boldsymbol{\Phi}_i$ via Adam optimizer and set all negative weights to 0 for all $2 \leq i \leq m$.
4:     Update $\boldsymbol{\theta}$ by applying one step of ISTA Beck & Teboulle (2009) on Problem (5).
5:     Return features corresponding to non-zero elements in $\boldsymbol{\theta}$.

---

## 2.2 LOW-RANK MMD METHOD

In many biological datasets especially microarray and sequencing data containing thousands of gene expressions of living organisms, the number of samples (e.g., humans or mice) is much less than the number of studied genes. As a result Problem (5) consists of many optimization parameters in the

high dimensional setting when the number of data points is very limited compared to the dimension of the data ($n \ll d$), In particular, the unknown covariance matrix $\Sigma$ has $\mathcal{O}(d^2)$ parameters to optimize. For instance, when $d = 10^4$ (or even larger for genomic datasets), $\Sigma$ contains $10^8$ variables, while only a few samples ($< 100$) is available. As mentioned earlier, a practical approach to avoid the challenging optimization procedure in (5) considers a multi-Gaussian distribution with a fixed randomly generated covariance matrix as the reference distribution. Although our simulation results show that even a fixed covariance matrix can beat the state-of-the-art approaches in the literature, the randomly generated covariance matrix can be arbitrarily far from the covariance matrix of the actual data distribution. Further, having small number of samples compared to the dimension of $\Sigma$, may cause the overfitting of the trained model.

To overcome the aforementioned issues, we exploit the fact that only a small proportion of input features are actually have strong correlations with the target variable in genomic datasets. In other words, in high-dimensional biological datasets (e.g., microarrays, gene expressions, etc), most pairs of features are almost independent. Thus, the ground-truth matrix is sparse, making the low-rank assumption on the covariance matrix of the data practically reasonable. Thus, we assume that the ground-truth covariance matrix is low-rank to reduce the parameters of the covariance matrix. This assumption reduces the number of optimization parameters significantly. Therefore, Problem (5) can be reformulated with an extra low-rank constraint on $\Sigma$ as follows:

$$\min_{\boldsymbol{\Phi}, \boldsymbol{\theta}, \Sigma} \quad \frac{1}{n} \sum_{i=1}^{n} \sum_{(\mathbf{x},y) \in \mathcal{D}_i} (\boldsymbol{\theta}^T \Phi_i(\mathbf{x}) - y)^2 + \lambda \|\boldsymbol{\theta}\|_1 + \mu \sum_{i=1}^{m} \text{MMD}\Big( \Phi_i(\mathcal{D}_i), \mathcal{N}(0, \Sigma) \Big)$$
$$\text{s.t.} \quad \text{Rank}(\Sigma) \leq s \tag{6}$$

Solving (6) is intractable due to the non-convexity of the rank constraint. Alternatively, since the covariance matrix must be symmetric and positive semi-definite, it can be represented as $\Sigma = A^T A$, where $A$ is a $s \times d$ matrix. In this case, $\text{Rank}(A^T A) \leq s$. Therefore, Problem (6) can be alternatively formulated as:

$$\min_{\boldsymbol{\Phi}, \boldsymbol{\theta}, A} \quad \frac{1}{n} \sum_{i=1}^{n} \sum_{(\mathbf{x},y) \in \mathcal{D}_i} \ell\Big( h_{\boldsymbol{\theta}}\big(\boldsymbol{\Phi}_i(\mathbf{x})\big), \boldsymbol{\Phi}_i(y) \Big) + \lambda \|\boldsymbol{\theta}\|_1 + + \mu \sum_{i=1}^{m} \text{MMD}\Big( \Phi_i(\mathcal{D}_i), \mathcal{N}(0, A^T A) \Big). \tag{7}$$

The standard approach for generating a zero-mean multi-Gaussian distribution ($X \sim \mathcal{N}^{n \times d}(0, \Sigma)$) is by first generating a standard normal Gaussian distribution $Z \sim \mathcal{N}^{n \times d}(0, I^{d \times d})$ and then generating a random matrix $A^{d \times d}$. The resulting multi-Gaussian distribution follows a covariance matrix of $A^T A$. In the problem context, the matrix $A$ is the variable matrix that needs to be trained.

$A$ can be generated with dimension $s \times d$ where $s \ll d$. Correspondingly, $Z \sim \mathcal{N}^{n \times s}(0, I^{s \times s})$ and then $X = AZ$. Consequently, the optimization problem is modified to include the low-rank matrix as follows:

$$\min_{\boldsymbol{\Phi}, \boldsymbol{\theta}, \boldsymbol{A}} \quad \frac{1}{n} \sum_{i=1}^{n} \sum_{(\mathbf{x},y) \in \mathcal{D}_i} \ell\Big( h_{\boldsymbol{\theta}}\big(\boldsymbol{\Phi}_i(\mathbf{x})\big), \boldsymbol{\Phi}_i(y) \Big) + \mu \sum_{i=1}^{m} \text{MMD}\Big( \Phi_i(\mathcal{D}_i), AZ \Big) + \lambda \|\boldsymbol{\theta}\|_1.$$

This modified approach is referred to as the "Low-Rank MMD" method, which is implemented and evaluated alongside Problem (5). The results, shown in Table 1, indicate that both the low-rank MMD method and original MMD methods perform well in two of the four scenarios. However, the low-rank method offers additional advantages and potential for further exploration. One of the key benefits of the low-rank approach is its increased explainability. By incorporating a low-rank matrix, the model becomes more interpretable, allowing a better understanding of the underlying factors influencing the data. Furthermore, the low-rank method demonstrates greater adaptability as the number of samples changes.

In particular, as we vary the low-rank parameter $s$ according to the number of samples, the low-rank model exhibits enhanced performance. This flexibility allows the model to effectively capture the underlying patterns and dependencies in the data, resulting in improved predictive power. By adjusting the low-rank parameter dynamically, the low-rank method can leverage the available information and adapt to different dataset sizes.

## 2.3 RANKING METHOD

An alternative, less computationally intense approach to the MMD-based method is to do prediction by relying on features that are invariant under batch effect transformations $f_1, \ldots, f_m$. In particular, since these transformations are monotonically increasing, **they do not change the order of entries in any row of the data matrix.** Thus, the order statistic in each row is unchanged, which means the row-wise order statistics is invariant under the batch effects. After applying the order statistics, one can perform Lasso regression or other feature selection algorithms on the transformed dataset. Despite its simplicity, using the order statistics instead of the original data can lead to information loss, as it only considers the orders not the magnitude of data. While the simple Ranking methods beats several state-of-the-art approaches in the literature based on our experiments, the method based on joint removal of batch effect and feature selection through Lasso regression is dominantly better than the ranking method in all synthetic and real data scenarios.

---

**Algorithm 2** Feature Selection with Ranking-based Method
---
1: Convert original data $(X, Y)$ to ranking data $\mathbf{R}(X, Y) = (X', Y')$ per row;
2: Apply Lasso on $\mathbf{R}(X, Y)$ and select corresponding features.

---

## 3  NUMERICAL EXPERIMENTS

In this section, we evaluate the performance of MMD-based and ranking-based methods on simulated datasets. To do this, we first consider the feature selection tasks in the presence of distributional shifts. For this task, we measure performance using the $\mathcal{F}1$ score which is defined as the harmonic mean of recall and precision. Moreover, to evaluate the effectiveness of the methods in removing batch effects, we visualize the data batches before and after the transformation.

### 3.1  SIMULATED DATA

To evaluate the performance of our proposed methods against state-of-the-art baselines in the literature, we generate datasets with different number of batches $m$ and the number of data points $n$ in each batch. To this end, we generate $mn$ data points with dimension $d = 100$. Each batch follows a normal distribution with a randomly assigned chosen mean and covariance. The target variable $y$ is a linear function of the input data plus a mean zero normal noise ($y = \mathbf{x}^T \beta^* + \epsilon$). To induce sparsity, each entry of $\beta^*$ is set to zero with the probability of $90\%$. To add batch effects to batch $k$, we transform the dimension $j$ in data point $i$ in batch $k$ as follows:

$$x'_{ij} = a_k x_{ij}^{\frac{5}{3}} + b_k x_{ij} + c_k + \epsilon_k$$

where $a_k$, $b_k$ and $c_k$ are randomly generated positive numbers for batch $k$ and $\epsilon_k$ is Gaussian noise. We vary the number of data points and data batches to evaluate the effectiveness of each approach in different scenarios. Besides MMD-based and ranking-based methods proposed in this paper, we evaluate several state-of-the-art approaches including CombatSeq (Zhang et al., 2020), Limma (Smyth, 2005), zero-mean-unit-variance, and PCA (Leek et al., 2010). We also compare outcomes when we apply Lasso on the datasets without moving batch effects. Table 1 reports the $\mathcal{F}1$ score for the aforementioned approaches in four different scenarios.

| **(m, n)** | S1 (5,10) | S2 (50,10) | S3 (5,100) | S4 (50,100) |
|---|---|---|---|---|
| Combat-Seq | 0.424 | 0.313 | 0.444 | 0.759 |
| Limma | 0.077 | 0.109 | 0.217 | 0.238 |
| PCA | 0.143 | 0,089 | 0.228 | 0.238 |
| Zero-Mean Unit-Variance | 0.061 | 0.204 | 0.231 | 0.16 |
| Original Data | 0.381 | 0.145 | 0.289 | 0.289 |
| Ranking | 0.444 | 0.095 | 0.214 | 0.214 |
| Shaham | 0.326 | 0.143 | 0.289 | 0.297 |
| MMD | **0.410** | **0.727** | **0.880** | **0.857** |
| Low-Rank MMD | **0.537** | **0.400** | **0.857** | **0.909** |

Table 1: $\mathcal{F}1$ Scores for Different Methods and Scenarios

## 3.2 HYPER-PARAMETER TUNING

The selection of suitable hyper-parameters in problem (5) significantly impacts the overall performance. However, determining optimal values for $\lambda$ and $\theta$ poses a considerable challenge. In realistic datasets, the true coefficient vector $\beta$ is unknown, making it difficult to assess the final results accurately. Consequently, an alternative metric is needed to replace the $\mathcal{F}1$ score for conducting cross-validation.

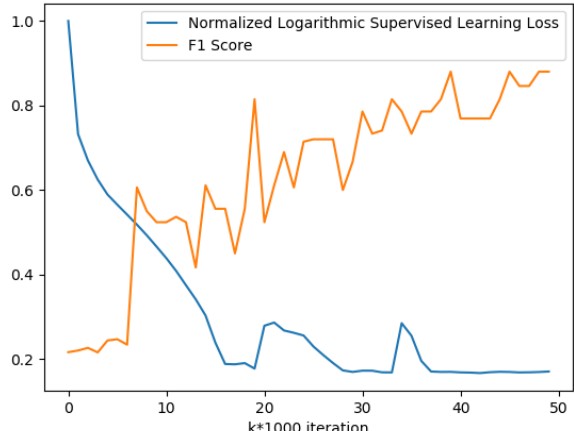

Figure 2: $\mathcal{F}1$ Score Versus Normalized Logarithmic Supervised Learning Loss

One potential indicator is the supervised learning loss. We calculate the $\mathcal{F}1$ score at regular intervals, typically every 1000 iterations, while simultaneously recording the logarithmic supervised learning loss. To facilitate comparison, we normalize the loss by dividing it by the maximum logarithmic supervised learning loss. Figure 2 demonstrates that as the supervised learning loss decreases, the $\mathcal{F}1$ score tends to increase.

## 3.3 INTERPRETING THE RESULTS

From Table 1, it is evident that the MMD-based approach performs significantly better than other methods by a large margin in Scenarios 2 to 4. In Scenario 1, Combat-Seq works slightly better. This can be attributed to the requirement in the MMD method that there be an adequate number of samples to obtain the optimal transformations (modeled by two-layer neural networks) effectively. Conversely, other benchmarks perform even worse than applying Lasso on the original datasets. This suggests that these approaches may alter the underlying information within the datasets. Additionally, the ranking method does not effectively select the correct features, potentially leading to the loss of crucial information.

## 3.4 CONVERGENCE OF THE TRAINING APPROACH

Based on Equation (5), the supervised learning loss, MMD loss and $\mathcal{L}1$ norm are expected to decrease during the training process as we minimize the objective. Figure 3 plots the training process for scenario 3, the upper left is the logarithmic prediction loss or supervised training loss, the upper right is the $\mathcal{L}1$ norm of $\theta$, the lower left is the sum of MMD between transformed data and reference data, the lower right is the objective value in (5). We can see overall the loss shows a declining trend. Small MMD loss indicates the batch effects are corrected, and the transformed data's distributions are close to that of the reference data. Figure 3 shows the MMD between each pair of datasets before and after transformation in scenario 3. The diagonals are black (MMD=0) because the statistical distance between a dataset and itself is 0.

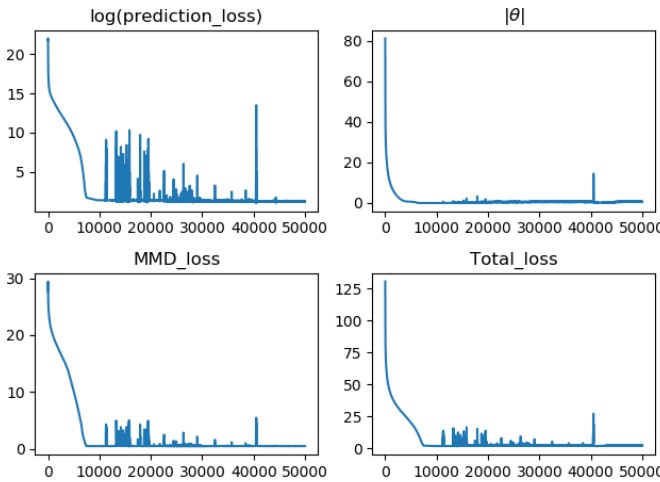

Figure 3: Training Loss

Non-diagonal values in Figure 4 (left) are roughly between 1 and 2 whereas the post-transformation values (right) are smaller than 0.1, showing that the batch effects have been corrected. In Appendix A, we further discuss the convergence behavior and quality of the transformed datasets obtained by our MMD-based approach. All codes and outcomes are publicly available at `https://anonymous.4open.science/r/Feature-Selection-in-the-Presence-of-Monotone-Batch-Effects-33CC/`.

Drawing inspiration from this observation, we can utilize cross-validation to identify the optimal values of $\theta$ and $\lambda$ that minimize the converged supervised learning loss for test datasets.

In addition, we have observed that when each term in equation 5, along with its corresponding coefficients, possesses similar magnitudes, the overall performance improves. This phenomenon has also been mentioned in Ouyang & Key (2021).

The idea behind this approach involves training the model with random values for $\lambda$ and $\theta$. Once the training loss converges, the values of $\lambda$ and $\theta$ are reset such that the terms $\frac{1}{n}\sum_{i=1}^{n}\sum_{(\mathbf{x},y)\in\mathcal{D}_i}\ell\left(h_{\boldsymbol{\theta}}\big(\boldsymbol{\Phi}_i(\mathbf{x})\big),\boldsymbol{\Phi}_i(y)\right)$ (supervised training loss), $\mu\sum_{i=1}^{m}\mathrm{MMD}\left(\Phi_i(\mathcal{D}_i),\mathcal{N}(0,\Sigma)\right)$ (MMD loss), and $\lambda\|\boldsymbol{\theta}\|_1$ ($\ell_1$ norm) are approximately equal. For instance, if the supervised training loss is 1000, the MMD loss is 0.1, and the $L_1$ norm is 10, we can set $\mu = 10^4$ and $\theta = 100$. This process is repeated until no further updates are needed for $\lambda$ and $\theta$. This method assists in determining suitable hyper-parameters and adjusting step sizes accordingly.

Subsequently, cross-validation is employed to search for the values of $\theta$ and $\lambda$ that minimize the supervised learning loss on the test dataset. The empirical settings for $\lambda$ and $\theta$ can also help narrow down the search range during cross-validation implementation.

## 3.5 GENOMIC DATA EXPERIMENTS

Batch effects also appear in datasets in biological applications, particularly in genomic data where a small number of samples may be pooled together for analysis from different laboratories. We therefore demonstrate our approach on an analysis of Srebp-1c expression in liver cells, which is a gene implicated in liver disease. Identifying genes with expressions correlated to Srebp-1c may help us narrow down which other genes should be studied to more fully elucidate gene pathways in liver disease.

To identify genes potentially correlated with Srebp-1c, we gathered 15 publicly available GEO datasets, originating from diverse environments. Our dataset collection includes the following: GSE149863, GSE94593, GSE162249, GSE138778, GSE137345, GSE162276, GSE163728, GSE163652, GSE151358, GSE157798, GSE140994, GSE156918, GSE162322, GSE162324, and GSE166347. We observed a significant batch effect within these datasets. For instance, the expression

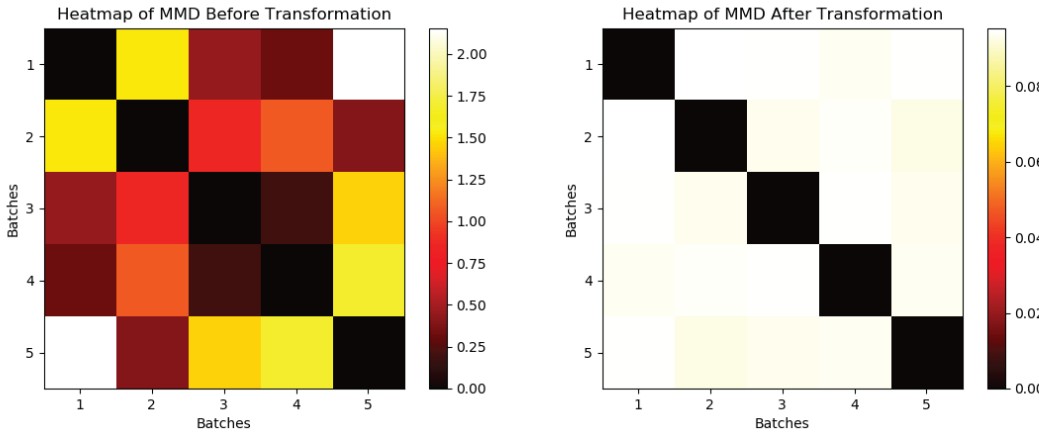

Figure 4: Heat-map of MMD before and after transformation

values in GSE137345 exhibit magnitudes below 10, considerably smaller than those in GSE156918, where half of the counts were above 1,000. Our proposed approach, which removes batch effect, is therefore well-suited for the task of identifying potential genes that are correlated with Srebp-1c.

To assess the performance of our method, we can compute the F1 score calculated by comparing the genes obtained through our approach with a list of known correlated genes. We do not have the ground truth when conducting genomic data analysis. However, some prior biological experiments have explored potential genes correlated with Srebp-1c (as referenced in Seo et al. (2009)). These experiments provide a list of genes known to be true correlated genes, serving as a benchmark. We use the F1 score as it provides a robust evaluation metric that balances precision and recall.

The highest F1 score attained in our analysis was 3 times higher than that of the F1 score using the direct Lasso method. This process identified 395 potential genes out of about 10,000 genes; 15 of them overlap with the 395 genes listed in the biological publication by Seo et al. Seo et al. (2009), which provides genes with biological connections with Srebp-1c. This suggests that this process may be useful for identifying other genes of biological interest.

Another way to identify correlated genes is by using the frequency of genes occurring among experiments with different hyperparameter settings. The higher the frequency, the more robust the relationship between a gene and the target gene will likely be. In our analysis, several genes appeared with high frequency, including Ubtd2, CD36, and Defb1. CD36 appeared 91 times among 100 experiments and, as confirmed by existing biological literature, it is known to correlate with Srebp-1cZeng et al. (2022). For other genes, additional biological experiments are needed to further explore their potential associations.

## 4   CONCLUSION

We proposed a joint optimization framework for feature selection using Lasso and removing batch effects by matching the distributions of datasets using MMD. Aside from feature selection, the method can be used as an effective tool to remove batch effects in a pre-processing task. Numerical experiments on different scenarios demonstrate a significant improvement in the performance of this approach compared to other state-of-the-art methods. Moreover, the results of genomic datasets conducted by MMD method are supported by existing biological literature, which demonstrates the practicality of our approach.

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

# A  VALIDATE MULTI-VARIATE GAUSSIAN DISTRIBUTION

In Figure 4 (right), we can observe that the Maximum Mean Discrepancy (MMD) values of the transformed datasets are not equal to $0$. We consider this to be a reasonable outcome because the reference dataset is generated randomly and is expected to differ from the true underlying distribution. The goal is for the transformed data to be close to a multivariate Gaussian distribution rather than an exact match to the reference data. If the MMD in Figure 4 were $0$, it would indicate overfitting. Figure 5 displays the histograms and overlays the corresponding Gaussian curves for the original data and the transformed data achieved through linear mapping with the random vector $\mathbf{a}$. Notably, after the transformation, the data exhibits a clear Gaussian distribution, unlike the data prior to the transformation.

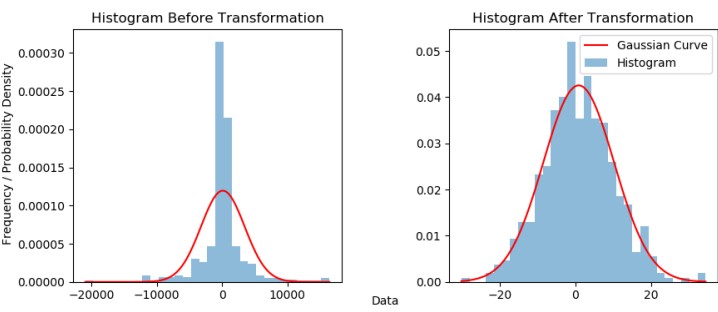

Figure 5: Histogram for Validating Gaussian Distribution

The transformed data does not fully converge to the reference data due to the design of the optimization problem 5. In Problem 5, we conceptualize minimizing the MMD as the main objective and the Lasso as a regularization term. Since the underlying distribution follows a sparse linear relationship, forcing the transformed data to be exactly the same as the reference data would lead to large supervised loss and $\mathcal{L}_1$ norm. Hence, appropriate tuning of hyper-parameters $\lambda$ and $\mu$ becomes crucial in achieving good performance.

