# OpenReview forum: "Feature Selection in the Presence of Monotone Batch Effects"
_ICLR.cc/2024/Conference — ICLR 2024 Conference Withdrawn Submission_

### Official Review · Reviewer_p82K · 2023-10-31

**Soundness:** 2 fair
**Presentation:** 1 poor
**Contribution:** 1 poor
**Rating:** 3
**Confidence:** 4

**Summary:**

The authors present a feature selection method for data that includes several datasets with batch effect. Such batch effect is typically evident in biological datasets that are collected on different days, labs, or measurement technologies. They assume that the batch effect is monotone and propose a method for simultaneous feature selection and batch effect removal. The main idea is to learn the monotone batch effect removal transformation by restricting the weights of a neural network and minimizing a maximum mean discrepancy loss between the distributions of different datasets. To perform feature selection, they exploit a LASSO-like L1 regularization. The method is evaluated on a synthetic and real example.

**Strengths:**

The problem of feature selection is important and has numerous applications. I believe that the batch effect setting is interesting and realistic, therefore, such a method could be valuable for the community.

**Weaknesses:**

The paper is not well written, and many parts are not presented in a clear way.
The notations and conventions are not consistent, and there are many mistakes/typos.
The algorithmic contribution is a combination of an MMD loss, and an L1 norm; while the combination of existing ideas could be considered somewhat novel, I find the specific choices of MMD and L1 not very up-to-date with the SOTA of either of the fields ( feature selection or batch effect removal).
There are more advanced schemes that enable nonlinear feature selection, for instance.
The experimental part is also pretty weak and only includes one synthetic example and one real-world data.
For the synthetic example, they only evaluate one underdetermined setting (m,n)=(5,10), and in this example, the method does not provide a significant advantage. Also, for this synthetic experiment, they do not provide the regression results of all models.
For the real dataset, there is no benchmark in they only compare to no batch effect removal without providing the actual F1 score (which, as they wrote, is actually ill-posed).

**Questions:**

The initials of lasso are never defined also it is written in three different ways in the paper: LASSO, Lasso, lasso
Many methods (and acronyms) are presented in the abstract without citation; that’s not a proper way to mention other work.

How is UMAP be used for batch effect removal? It is quite different than PCA; you can not remove the information and project back. It might be useful only for visualization purposes, not batch effect removal per se.

Many missing citations for batch effect removal:

Korsunsky, Ilya, et al. "Fast, sensitive and accurate integration of single-cell data with Harmony." Nature methods 16.12 (2019): 1289-1296.

Lopez, Romain, et al. "Deep generative modeling for single-cell transcriptomics." Nature methods 15.12 (2018): 1053-1058.

Many missing citations for feature selection:

Yamada, Yutaro, et al. "Feature selection using stochastic gates." International Conference on Machine Learning. PMLR, 2020.

Muthukrishnan, Ramakrishnan, and R. Rohini. "LASSO: A feature selection technique in predictive modeling for machine learning." 2016 IEEE international conference on advances in computer applications (ICACA). IEEE, 2016.


In section 2, “the goal of batch effect…” isn’t the goal to learn the inverse of these functions?

How does the Gaussian assumption hold for actual biological data, for example, scRNA seq, which is typically modeled by other distributions?

The dimensions of most terms in the paper are not defined; for example \Phi…

The NN function \Phi, is it applied to each feature separately? or does it learn feature interactions like standard NNs? This is not clear, specifically because the parameter \theta is multiplied by the output of this NN. So if it is actually performing FS \Phi has to be an element-wise function (so not a fully connected NN).

On page 5, many statements are incorrect, for example, that most pairs of genes are almost independent. In scRNA seq, many large groups of features are correlated.
 The transition from eq.6 to 7 is unclear. Suddenly there is $h$ $\ell$ there is also twice +

L1 is written in many ways in the paper.
F1 is not defined. I understand that this is F1 of feature selection, but this is not explained.

The convergence analysis is not really informative and is more appropriate in an appendix.

Use the same color scale in Figure 4 otherwise, it's confusing.

Why not try the method for cell classification with batch effect, then, the method could be evaluated in terms of a real biological task?

---

### Official Review · Reviewer_xAdr · 2023-10-31

**Soundness:** 3 good
**Presentation:** 2 fair
**Contribution:** 2 fair
**Rating:** 3
**Confidence:** 3

**Summary:**

The paper studies how to gather multiple data sets, remove the batch effect that each of them may have and perform feature selection on the unified dataset to optimize the prediction task. The paper is mostly experimental. The mathematical formulations are presented to define the problem, and motivate the empirical approach taken by the paper. No proof is provided to support the algorithmic decisions they made and the way they formulated the objective function.

 The experimental results are mainly on a simulated instance of a few or  a limited number of batches (5 or 50 datasets) with 10 or 100 points in each dataset. The distribution of each data set seems to be an independent Gaussian distribution which is surprising. There is no reason to believe that there is a reasonable unifying distribution (ground truth solution) in this synthesized dataset. So the main purpose of this experiment could be mostly as a test case that their algorithms are running as intended.

They also have experiments on the Genomic dataset but they say there is no ground truth for this dataset. They do some comparisons with the Lasso method and claim to have a higher F1 score. As part of this comparison they mention:
“ This process identified 395 potential genes out of about 10,000 genes; 15 of them overlap with the 395 genes listed in the biological publication by Seo et al. Seo et al. (2009), which provides genes with biological connections with Srebp-1c.”

It is not clear how significant finding 15 out of the 395 genes is.

**Strengths:**

They provide some reasonable experimental ideas to explore.

**Weaknesses:**

The paper is mostly experimental. The mathematical formulations are presented to define the problem, and motivate the empirical approach taken by the paper. No proof is provided to support the algorithmic decisions they made and the way they formulated the objective function.

 The experimental results are mainly on a simulated instance of a few or  a limited number of batches (5 or 50 datasets) with 10 or 100 points in each dataset. The distribution of each data set seems to be an independent Gaussian distribution which is surprising. There is no reason to believe that there is a reasonable unifying distribution (ground truth solution) in this synthesized dataset. So the main purpose of this experiment could be mostly as a test case that their algorithms are running as intended.

They also have experiments on the Genomic dataset but they say there is no ground truth for this dataset. They do some comparisons with the Lasso method and claim to have a higher F1 score. As part of this comparison they mention:
“ This process identified 395 potential genes out of about 10,000 genes; 15 of them overlap with the 395 genes listed in the biological publication by Seo et al. Seo et al. (2009), which provides genes with biological connections with Srebp-1c.”

It is not clear how significant finding 15 out of the 395 genes is.

**Questions:**

The experimental results are mainly on a simulated instance of a few or  a limited number of batches (5 or 50 datasets) with 10 or 100 points in each dataset. The distribution of each data set seems to be an independent Gaussian distribution which is surprising. There is no reason to believe that there is a reasonable unifying distribution (ground truth solution) in this synthesized dataset. So the main purpose of this experiment could be mostly as a test case that their algorithms are running as intended.

They also have experiments on the Genomic dataset but they say there is no ground truth for this dataset. They do some comparisons with the Lasso method and claim to have a higher F1 score. As part of this comparison they mention:
“ This process identified 395 potential genes out of about 10,000 genes; 15 of them overlap with the 395 genes listed in the biological publication by Seo et al. Seo et al. (2009), which provides genes with biological connections with Srebp-1c.”

It is not clear how significant finding 15 out of the 395 genes is.



More detailed comments:
In problem formulation, they mention functions f_i are monotone. These are functions that are applied on distributions, each represented by a collection of vectors of features. What does it mean for these functions to be monotone? Their inputs do not adhere to any total ordering.

E_{z,z’} can be understood from the context but I suggest defining it explicitly.

Page 4: they mention input features x^* follow a multivariate Gaussian distribution. Is this a standard assumption?

The transformation functions should be bijective. Could you provide an explanation on why the ReLu transformation is bijective?

Page 5: “Although our simulation results show that even a fixed covariance matrix can beat the state-of-the-art approaches in the literature, the randomly generated covariance matrix can be arbitrarily far from the covariance matrix of the actual data distribution.”
You claim that you beat the state of the art with a fixed matrix but these two methods do not seem comparable. When you generate a random fixed matrix, your objective is to minimize the sum of distances to this generated matrix IIUC. Whereas in the other methods you have other objectives to optimize.

Page 6, since the notion of monotonicity for f_i is not defined properly, the claim about order statistics of rows are also not clear.

Subsection 3.1: It seems that you generate a separate random normal distribution for each of the m datasets. If this is not the case, you need to explain it in more detail.

Page 7, last paragraph: Figure 3 shows the MMD … → Figure 4 …

---

### Official Review · Reviewer_6Rhw · 2023-11-06

**Soundness:** 3 good
**Presentation:** 2 fair
**Contribution:** 2 fair
**Rating:** 3
**Confidence:** 4

**Summary:**

The paper focuses on variable selection in high-dimensional regression, in the presence of monotone batch effects resulting from aggregation of multiple data sets. The authors introduce a new methodology to simultaneously select features and remove batch effects, by optimizing a cost function combining a Lasso regression loss and a distribution matching loss based on maximum mean discrepancy. In addition, they introduce an extended version of the method which accounts for simultaneous low-rank modeling, which can be useful especially in genomics applications.

**Strengths:**

- The focus of the paper is an important problem encountered in applied data analysis, particularly in health care applications. Thus, the methodology could have a significant impact.
- The simulated experiments are quite thorough with comparisons with 6 state of the art competitors, and the proposed methods clearly outperform competitors on the metric used.

**Weaknesses:**

- The paper has flaws in the presentation, and looks like it has not been properly proof-read. For instance, some sentences of the literature review are written twice in different paragraphs. ("An alternative approach
is to apply a clustering algorithm on the data and remove the batch effect iteratively through the
clustering procedure Li et al. (2020); Fang et al. (2021); Lakkis et al. (2021). More specifically, each
data batch is considered a cluster, and the batch effect is viewed as the between-cluster variances Fang
et al. (2021). " p.1 and 2).
- The mathematical formulation is not always very clear. For instance, p.3 it is not clear how the monotonicity of the functions f_i is defined, as they are multidimensional. See also questions section below.
- The real data experiments are quite limited for a methodological paper. In particular, the method is not compared to competitors on the real data experiments, which is a shame as it limits the interpretation and understanding of the paper's impact.

**Questions:**

- Could you provide a bit more details on why "Such assumptions may be invalidated when the batch effect removal
procedure does not account for the downstream feature selection procedure." (p.2) ? This is not obvious while reading and is quite important to understand what could be the impact of the paper.
- The literature review on variable selection is not very thorough, as only multiple testing and Lasso are referred to, which are important but not very recent. Could you perhaps extend a little bit this paragraph and explain whether more recent work (e.g. SLOPE https://arxiv.org/abs/1407.3824 or post-selection inference https://www.stat.cmu.edu/~ryantibs/statml/lectures/Lee-Sun-Sun-Taylor.pdf) are also sensitive to batch effects ?
-  p. 3 Are the functions f_i required to operate pointwise ineach data set ? Or could there be interactions between data points ? If so how do you define monotonicity ?

---

### Author Response · Authors · 2023-11-22
**Response to Reviewers**

We thank the reviewers for their constructive feedback. After reading the reviews, we believe there was major confusion about the contributions and the problem statement. To clarify the problem statement and to highlight the contributions, we decided to withdraw and make major changes to the paper.